# X10 expansion microscopy enables 25-nm resolution on conventional microscopes

Sven Truckenbrodt[1,2,*,†], Manuel Maidorn[1,2], Dagmar Crzan[1], Hanna Wildhagen[1], Selda Kabatas[1] & Silvio O Rizzoli[1,**] (iD)

## Abstract

**Expansion microscopy is a recently introduced imaging technique that achieves super-resolution through physically expanding the specimen by ~4×, after embedding into a swellable gel. The resolution attained is, correspondingly, approximately fourfold better than the diffraction limit, or ~70 nm. This is a major improvement over conventional microscopy, but still lags behind modern STED or STORM setups, whose resolution can reach 20–30 nm. We addressed this issue here by introducing an improved gel recipe that enables an expansion factor of ~10× in each dimension, which corresponds to an expansion of the sample volume by more than 1,000-fold. Our protocol, which we termed X10 microscopy, achieves a resolution of 25–30 nm on conventional epifluorescence microscopes. X10 provides multi-color images similar or even superior to those produced with more challenging methods, such as STED, STORM, and iterative expansion microscopy (iExM). X10 is therefore the cheapest and easiest option for high-quality super-resolution imaging currently available. X10 should be usable in any laboratory, irrespective of the machinery owned or of the technical knowledge.**

**Keywords** expansion microscopy; nanoscopy; STED; STORM; super-resolution
**Subject Category** Methods & Resources

## Introduction

The resolution of fluorescence microscopes has been limited by the diffraction barrier to approximately half of the wavelength of the imaging light (in practice, 200–350 nm). This barrier has been lifted by several microscopy concepts, for example, by using patterned light beams to determine the coordinates from which fluorophores are permitted to emit, as in the stimulated emission depletion (STED) family [1,2], or by determining the positions of single fluorophores that emit randomly, as in photo-activated localization microscopy (PALM) [3], stochastic optical reconstruction microscopy (STORM and dSTORM) [4,5], or ground state depletion microscopy followed by individual molecule return (GSDIM) [6] Although such technologies have been applied to biology for more than a decade, their general impact on biomedical research is still relatively limited. This is mainly due to the fact that accurate super-resolution is still available only for selected laboratories that are familiar with the different tools, are able to apply the appropriate analysis routines, and/or possess the often highly expensive machinery required.

The ideal super-resolution tool for the general biologist needs to be easy to implement, without specialized equipment and without the need for complex imaging analysis. At the same time, such a technique would need to be highly reliable and should be easy to apply to multiple color channels simultaneously. The expected resolution should be at least on the size scale of the labeling probes used. This would be ~20–30 nm for normal immunostaining experiments, since these rely on identifying the epitopes via primary antibodies that are later detected through secondary antibodies, each of which is ~10–15 nm in size. Expansion microscopy, a technique introduced by the Boyden laboratory [7–10], is an important step in this direction. Expansion microscopy entails that the sample of interest is first fixed, permeabilized, and immunostained and is then embedded in polyelectrolyte gels, which expand strongly when dialyzed in water. To ensure that no disruption of the sample aspect ratio occurs, the sample is digested using proteases after embedding, but before the expansion step. The fluorophores, which are covalently bound to the gel, thus maintain their relative positions, although they are now positioned a few-fold farther away from each other than in the initial sample. The preparation can then be imaged in a conventional microscope. This renders expansion microscopy the simplest approach, to date, that is able to produce super-resolution images. However, the initial implementations of this approach were performed with gels that expanded, on average, about fourfold, which resulted in lateral resolutions of ~70 nm, i.e. not as high as that of modern STED or STORM microscopes [7]. The only solution proposed so far to this problem has been to use

1 Institute for Neuro- and Sensory Physiology, Center for Biostructural Imaging of Neurodegeneration, Cluster of Excellence Nanoscale Microscopy and Molecular Physiology of the Brain, University Medical Center Göttingen, Göttingen, Germany
2 International Max Planck Research School for Molecular Biology, Göttingen, Germany
  *Corresponding author. Tel: +43 2243 9000 2063; E-mail: strucke@gwdg.de
  **Corresponding author. Tel: +49 551 39 5911; E-mail: srizzol@gwdg.de
  †Present address: Institute for Science and Technology Austria, Klosterneuburg, Austria

complex procedures consisting of multiple successive expansion steps (iterative expansion), which would require the embedding, expansion, re-embedding, and re-expansion of the sample.

We set out here to solve this problem, by generating a protocol that uses only one embedding and expansion step, but still obtains a resolution of the required value (20–30 nm), in multiple color channels, without any difficult techniques, tools, or analysis routines. Our protocol expands the sample by 10-fold, and we therefore termed it X10 microscopy. It achieves a resolution of 25–30 nm on conventional epifluorescence microscopes and does not even require confocal imaging for accurate nanoscale analyses. We compared X10 microscopy with state-of-the-art commercial implementations of both STED and STORM, and found it to be superior to both. Judging from the available literature, it is clear that self-built super-resolution microscopes, operated and optimized by specialists, could provide images that are superior to our X10 implementations on epifluorescence setups. In spite of this, the fact that X10 is the simplest and cheapest super-resolution technique currently available, with a resolution performance that is superior to what is available to the general biologist (i.e. the commercial implementations of these techniques), should render it the tool of choice for the implementation of super-resolution in the general biology laboratory.

## Results and Discussion

To obtain a resolution of 20–30 nm within a one-step expansion procedure, we generated a protocol that enables the use of a super-absorbent hydrogel designed for excellent mechanical sturdiness [11] for the expansion of biological samples. This gel uses *N,N*-dimethylacrylamide acid (DMAA) for generating polymer chains, which are crosslinked with sodium acrylate (SA) to produce a swellable gel matrix (Fig 1). The gelation–free radical polymerization reaction is catalyzed by potassium persulfate (KPS) and *N,N,N',N'*-tetramethylethylenediamine (TEMED; Fig 1A), and produces a gel that can expand > 10× in each dimension when placed in distilled water (Fig 1B and C). Protein retention in the gel is achieved via the previously described anchoring approach [8,9], by employing Acryloyl-X. This uses NHS ester chemistry to covalently attach to proteins, while a second reactive acrylamide group integrates into the polymerizing gel matrix.

The different steps of the gel formation and protein retention reactions were initially difficult to optimize and therefore required extensive testing and fine-tuning. Nevertheless, the final version of the protocol is trivially simple and is highly reproducible. We present the critical steps in red in Fig 1, and we have included a complete protocol in Materials and Methods. Briefly, the main issues are the following. First, the reactions are extremely fast, and therefore, low temperature and high speed of application are essential, unlike in the gels used for 4× expansion. Second, oxygen inhibits polymerization and therefore needs to be carefully eliminated by bubbling with $N_2$. This is a trivial procedure, which requires no specialized setup (other than a tube to conduct the $N_2$ gas from a pressured gas container into the reaction mixture). Third, the gelation is initially rapid (it only takes minutes for the initial hardening), but does not continue with the same speed, and therefore, care must be taken that the gel is allowed to polymerize for 6–24 h.

Should these steps be followed as described here, and the resulting gel formation is highly reproducible (Figs 2 and EV1). The maximum expansion factor we achieved with this approach was ~11.5× (Fig 2A and C; see also Materials and Methods), which results in images with an apparent lateral resolution of ~25–30 nm (predicted from Abbe's resolution limit; Figs 2A–C and 3), in which substantially more details are revealed (Fig 2A–C and Movie EV1).

The resulting technique is fully compatible with the use of common affinity probes, such as antibodies (Fig 2), since X10 requires no specially designed labeling tools, similar to recent improvements to the 4× expansion [8,9]. The distortions of the sample introduced by the gel during swelling are minimal (Fig 2D) and are virtually identical to those seen in 4× expansion microscopy [7,8,12]. We would like to note, however, that the extensive digestion required for X10 is incompatible with expansion microscopy protocols that preserve fluorescent proteins [9]. These protocols utilize a milder digestion that retains some fluorescent proteins. This milder digestion, however, does not allow X10 to retain the sample integrity at higher expansion factors (Fig EV1). Therefore, fluorescent proteins will be visualized in X10 only by immunostaining them. However, this is not a major difficulty, as antibodies are currently available for all major fluorescent proteins. We would also like to note that X10 once more highlights the need for new probes for super-resolution imaging, as conventional antibodies usually do not result in a continuous staining of microtubules, but in a pearls-on-a-string pattern (as visible in Fig 2). This artifact, which is due to incomplete epitope coverage through conventional antibodies [13,14], can be observed also in many published works using other super-resolution techniques, such as STED [14–18] and STORM [18–22] (see also Appendix Fig S1). Alternatively, highly optimized tubulin labeling protocols should be used to ensure optimal epitope coverage.

To verify the resolution of X10 experimentally, we relied on investigating peroxisomes, which are round organelles with dimensions of ~100–200 nm in neurons. We immunostained Pmp70, a protein of the peroxisome membrane (Fig 3), and we compared pre-expansion images with post-expansion images, as well as with STED and STORM images (Fig 3A; see Fig EV2 for a more detailed comparison and Movie EV2 for a *z*-stack through several peroxisomes). To determine the nominal resolution of X10, we drew line scans through the membranes of the peroxisomes (post-expansion), fitted them to Gaussian curves, and determined their full width at half maximum values (FWHM; Fig 3B). The resolution determined in this fashion fits the theoretical prediction from Abbe's resolution limit that we have stated above, being centered at $25.2 \pm 0.2$ nm (Fig 3C).

We have also simulated peroxisomes stained for Pmp70, taking into account the size and random orientation of the primary/secondary antibody complexes (Fig EV3; see Materials and Methods for details), and found that the measured resolution value fits closely to the one predicted by the simulations (22.8 nm, on 10,000 simulated peroxisomes). This level of resolution is usually only achieved in highly specialized applications of STED and STORM [23,24] or in iterative expansion microscopy (iExM) [25] and is bettered substantially only by a recently developed tool, MINFLUX microscopy [26]. When investigating the same protein in state-of-the-art commercial STED and STORM setups, the image quality

**Figure 1. X10 gel polymerization reactions.**

A  Primary TEMED, sulfate, and hydroxyl radicals are generated by redox initiation with KPS and TEMED [37].

B  Radical propagation occurs when the monomer (DMAA) [11,38], ionic co-monomer (SA) [11], and tissue-anchored acryloyl monomer react with the primary radicals. Besides linear growth of the resulting polymer, DMAA also cross-links after proton abstraction at the methylene group [11,38].

C  The radical chain grows by reacting with monomers and through radical transfer to monomers or other polymers to form a branched network. Critical steps are shown in red.

these techniques achieved was, at best, comparable to that of X10 (Figs 3A and EV2). We used the modeling approach to determine whether we could, theoretically, resolve the lumen of microtubules, but found that this is beyond the limits of X10, when implemented with epifluorescence microscopy (Appendix Fig S2), due to problems in the placement of antibodies across the expanded microtubule. Their large size effectively limits the level of detail that can be observed [25,27], which implies that the ~25-nm resolution is possibly the maximum useful resolution that can be achieved in expansion microscopy when using conventional primary/secondary antibody stainings and epifluorescence microscopy. The use of antibodies in X10 can indeed blur the original staining, resulting in a larger perceived object (Appendix Fig S3).

The X10 procedure can be used to achieve multi-color super-resolution imaging. We could easily resolve, for example, synaptic vesicle clusters in cultured hippocampal neurons, along with the pre-synaptic active zones and the post-synaptic densities (Fig 4A–C; Movies EV3 and EV4). This enabled us to measure the distance between the pre-synaptic active zone, identified by Bassoon, and the post-synaptic density, identified by Homer 1 (Fig 4D). We found this distance to be ~120–140 nm, very similar to what has been previously described for these proteins using STORM [24,28]. The overall organization of the Homer 1 and Bassoon immunostainings, in the form of loose clusters containing multiple areas of dense packing, is also very similar to what has been observed with advanced STORM [29]. This type of information could not be

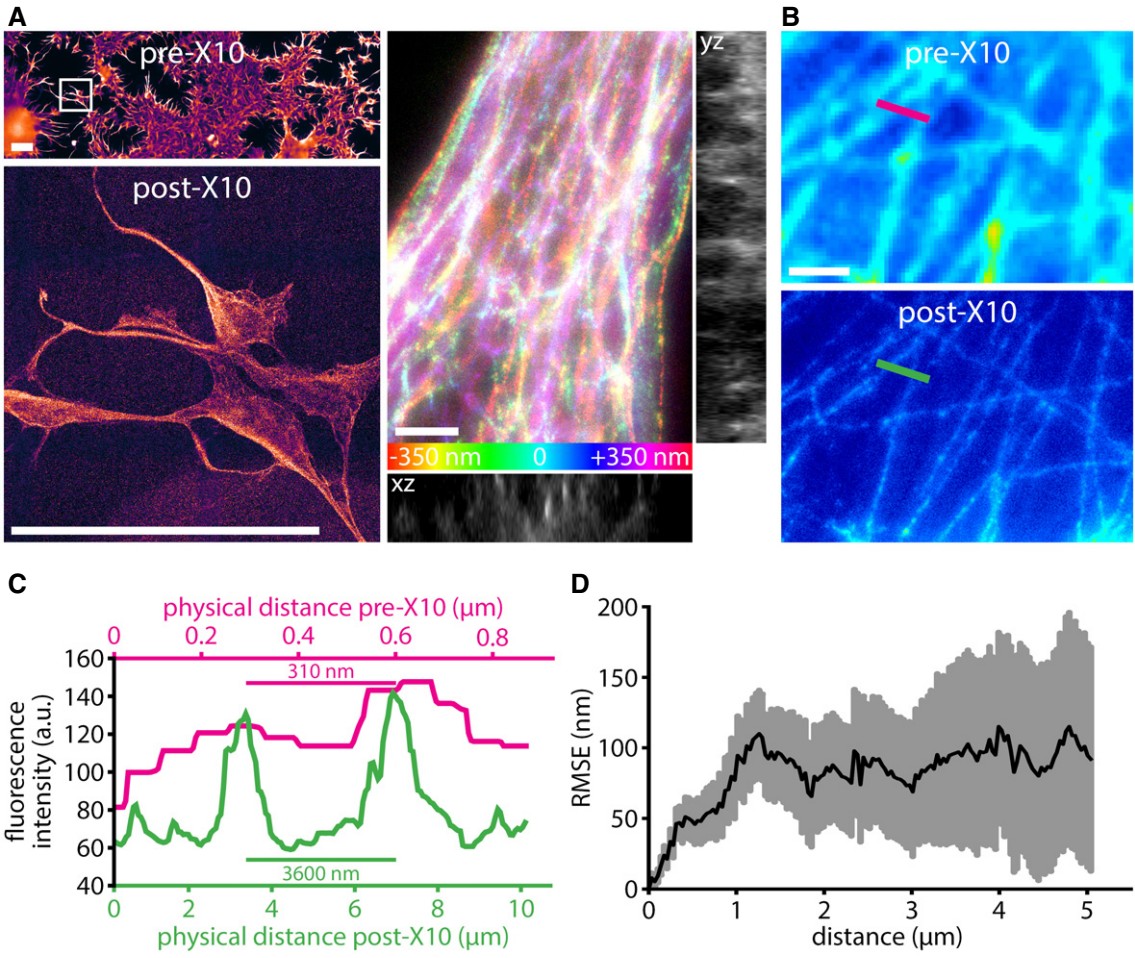

**Figure 2. X10 achieves super-resolution of biological samples on conventional epifluorescence microscopes.**

A The X10 gel is swellable to > 10× of its original size. The top panel on the left shows an overview image of COS7 cells stained for tubulin, before expansion. The bottom panel shows the cells framed in the top panel (white rectangle), after expansion. The images are to scale, demonstrating an expansion factor of 11.4× in this example. Scale bars: 100 μm (both panels). The post-expansion image is dimmer, as the fluorophores are diluted ~1,000-fold and therefore requires a longer acquisition time and a higher camera gain. The right panels reveal the 3D organization of the tubulin network in COS7 cells. The relative axial position of the fluorophores is visualized in a z-stack projection by color-coding (see scale at the bottom). Orthogonal views are given next to the z-stack projection (yz view across the midline, xz view along the bottom). A movie through this z-stack, including a rocking projection, is available in Movie EV1. Expansion factor: 11.4×. Scale bar: 1 μm.

B Comparison between pre-expansion resolution of tubulin imaging in COS7 cells (upper panel) and post-expansion resolution in the same sample (lower panel). Note that the images have not been processed to minimize distortions or to achieve a better correlation. Expansion factor: 11.5×. Scale bar: 1 μm.

C An exemplary measurement for the X10 expansion factor. A line scan was drawn over corresponding regions before and after expansion, as indicated in panel (B) by the colored lines.

D An analysis of the root mean square error (RMSE) of the distortions between aligned pre- and post-expansion images (see Materials and Methods for details; n = 34 automated measurements from four independent experiments).

obtained either with conventional microscopy or with classical 4× expansion microscopy (Appendix Fig S4). At the same time, we also analyzed, in a similar fashion, the pre-synaptic active zone proteins RIM1/2 and the post-synaptic marker PSD95, which should be separated by ~80 nm [24]. We found them to be separated by this distance also in X10 experiments (Fig 4E and F). This type of analysis can be performed also in confocal microscopy, but one does not necessarily obtain much information (Fig EV4), as the dim samples obtained by 1,000-fold volume expansion are not ideal for confocal imaging.

Overall, such examples demonstrate that X10 microscopy can reproduce results that were previously obtained only with highly specialized imaging tools. In addition, the ease with which multiple colors can be investigated in X10 is an advantage over previous localization microscopy methods, which have been typically limited to two-color channels in practice. Importantly, X10 microscopy can also be applied to thin tissue slices, where it provides the same resolution enhancement (Figs 5A–C and EV5; Appendix Fig S5; Movies EV5 and EV6). We have optimized X10 for brain slices, where this provides reliable measurements (Appendix Fig S5), but we would like to point out that extensive optimization will be required for other tissues or conditions. While we found X10 trivial to apply to cell cultures, this has not been the case for tissues, where optimal homogenization is essential. The difficulty in achieving such

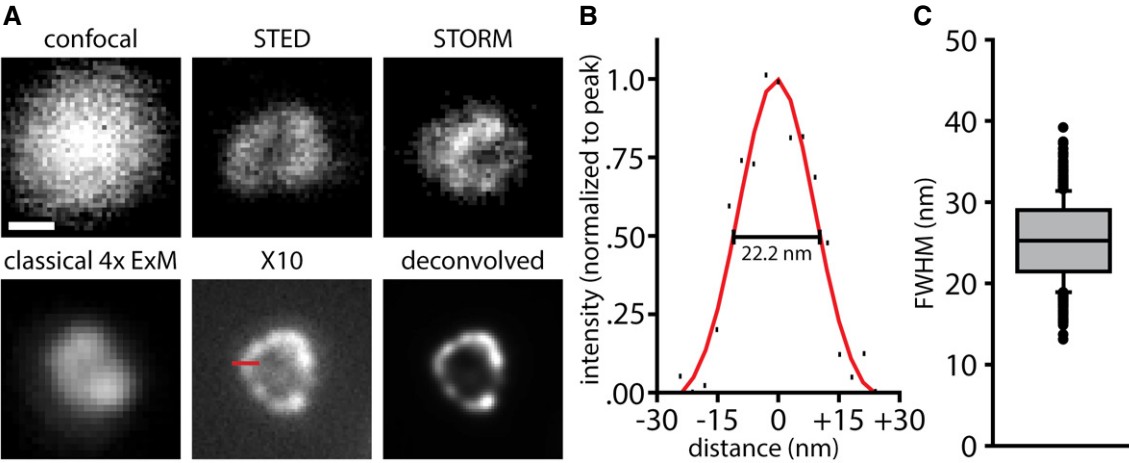

**Figure 3.  The resolution of X10 is ~25 nm.**

A  Immunostainings for the peroxisome membrane protein Pmp70 in neurons are shown. The first five panels show individual peroxisomes imaged with a confocal microscope before expansion, with a STED microscope before expansion, with a STORM microscope before expansion, with an epifluorescence microscope after classical 4× expansion microscopy, and with an epifluorescence microscope after X10 (without and with deconvolution). Expansion factors: 3.8× for classical 4× expansion microscopy and 9.5× for X10. Scale bar: 100 nm (applies to all panels). The red line in the X10 panel indicates a line scan over the peroxisome membrane (60 nm in length). See Fig EV2 for further examples.

B  The exemplary line scan from the X10 image in (A) is shown with a best Gaussian fit curve, with an indicated measurement of resolution as the full width at half maximum (FWHM).

C  A quantification of the average resolution, which is 25.2 ± 0.2 nm (*n* = 653 line scans across peroxisomes from two independent experiments). The data are represented as a box plot with median (horizontal line) and upper and lower quartile boundaries (box range), plus 1.5 times inter-quartile range (whiskers) and outliers (dots).

homogenization increases with the toughness of connective tissue in the sample, and a common result is fragmentation during expansion (e.g., Appendix Fig S6). One may nevertheless still obtain good imaging results for individual cells or tissue pieces, but optimal whole-tissue expansion is not as easily obtained as cell culture expansion. Two reactions may be especially optimized. First, the gel anchoring, in which Acryloyl-X is linked to amines in the proteins via NHS ester chemistry. This can be performed at basic pH values (for example in bicarbonate buffers, pH ~8.3), to increase its efficiency (Appendix Figure S6C and D). Second, the homogenization, which can be performed in buffers containing $Ca^{2+}$, to increase the efficiency of the proteinase K activity, or can be performed by other procedures such as autoclaving the tissue [9].

We therefore conclude that X10 microscopy provides the same resolution as reported in iterative expansion microscopy (iExM) [25]. The average resolution values are 25.2 nm for X10 and 25.8 nm for iExM. As indicated above, in iExM, the classical 4× gel is applied to the same sample multiple times in sequence, to achieve a multiplication of the expansion factors of the individual gels. This approach yields an expansion factor of up to ~16–20×, in two expansion steps. However, iExM is much more variable in its expansion factor than X10, as the multiplication of two 4× gels also results in a multiplication of the variability in their individual expansion factors, which usually spans from 3.5× to 4.5× [7–10,25]. At the same time, the iterative protocol of iExM requires additional time and effort to break the first gel and prepare the second gel, and is not compatible (at the moment) with the use of conventional off-the-shelf antibodies, but requires custom-made DNA-oligo-coupled antibodies [25]. This makes iExM much more complex, more time-consuming, and less precise than X10.

That said, it may be possible in the future to combine the X10 gel with the iExM principle to achieve lateral expansion factors of up to 100×, which would theoretically offer a resolution of up to 3 nm, and which should be useful, if probes smaller than antibodies are employed. As the antibodies place the fluorophores at a substantial distance from the epitopes, the additional image detail obtained with expansion factors > 10–12× comes at a disadvantage: The size of the measured structures no longer fits with the size of the actual objects measured, as shown by our simulations of microtubules (Appendix Fig S2) and by recent microtubule imaging results with iExM [25]. The microtubule lumen can be resolved at expansion factors beyond ~15× using antibodies, but the apparent wall-to-wall distance of microtubules exceeds the expected microtubule diameter by 20–25 nm, averaging ~50–55 nm both in our simulations and in published iExM results [25]. It is probably this factor that limits the effective resolution of iExM to ~25 nm, despite the larger expansion factor. X10, whose resolution is at the size of the probes, suffers less from this problem. Assuming that the Pmp70 epitopes surpass the peroxisome membrane by ~5 nm [30] and that the protein and antibody orientations are randomized after the permeabilization of the 6.8-nm-thick peroxisome membrane, the displacement induced by the antibodies is ~6 nm (Fig EV3).

X10 thus provides a toolset for cheap (< 2$ for all reagents used in one experiment, except antibodies) multi-color super-resolution imaging, which can be performed on widely available epifluorescence setups. This procedure provides a resolution that is superior to current combinations of conventional expansion microscopy and super-resolution, as published very recently using structured illumination [31], since this resulted in a resolution of ~30 nm, lower than

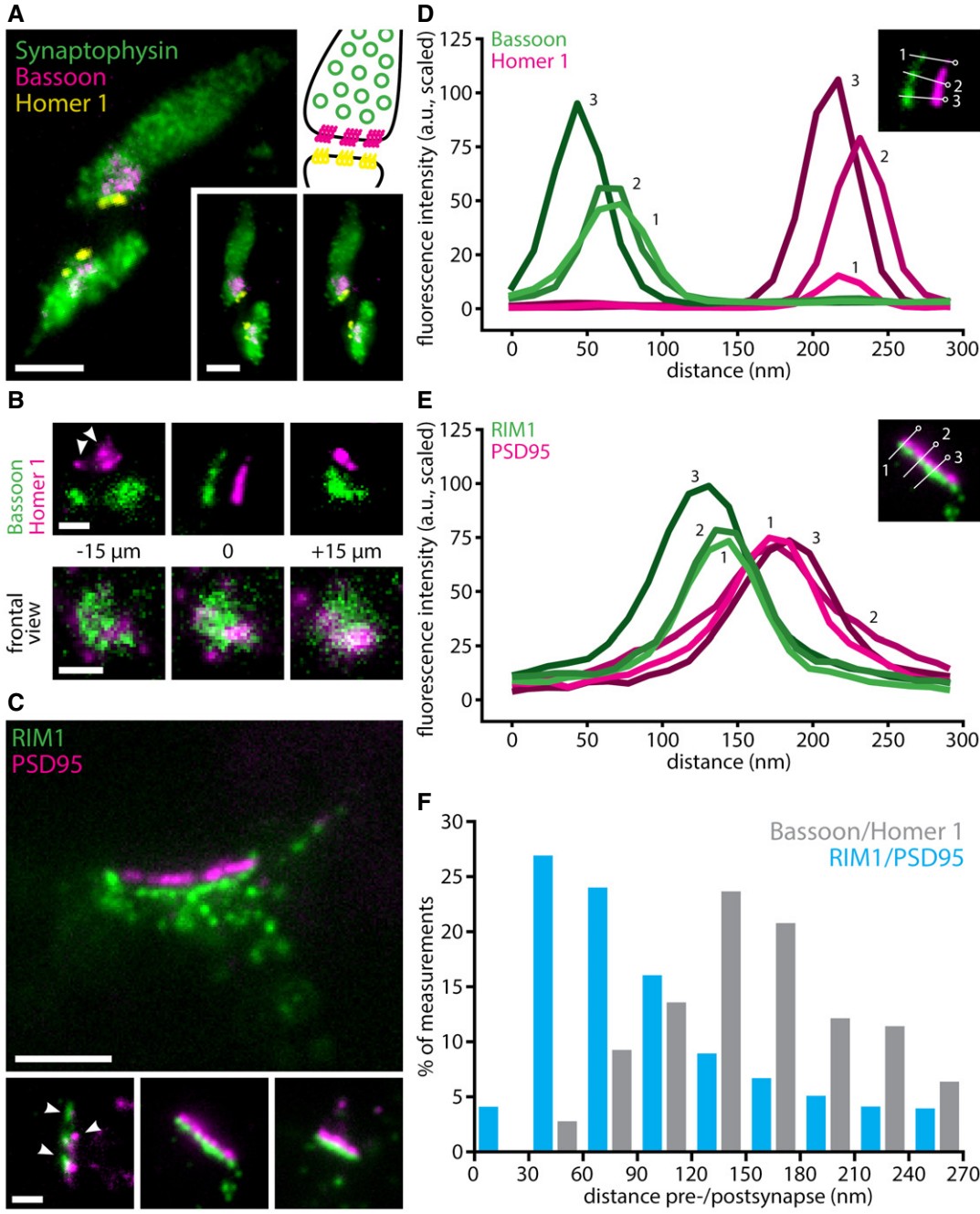

**Figure 4.  Multi-color imaging with X10 reveals synaptic ultrastructure in cell culture.**

A   Three-color imaging resolves synaptic vesicle clusters (identified by Synaptophysin), along with pre-synaptic active zones (identified by Bassoon) and post-synaptic densities (identified by Homer 1). The panel at the top right gives a schematic overview of the organization of a synapse, for orientation (colors as in the fluorescence images). The two panels on the bottom right provide a stereo view of the synapses. Expansion factor: 11.0×. Scale bars: 500 nm (both).

B   Upper panels: higher magnification images show the alignment of pre-synaptic active zones and post-synaptic densities, as well as the distance between them, in side view. Expansion factor: 11.0×. Scale bar: 200 nm. Lower panels: a z-stack through an additional synapse, in face view. Expansion factor: 11.0×. Scale bar: 200 nm.

C   Representative images of an immunostaining for pre-synaptic RIM1/2 and post-synaptic PSD95, two markers known to be more closely associated than Bassoon/Homer 1 [24]. Arrowheads indicate nanocolumns of aligned pre- and post-synaptic proteins. Expansion factor: 10.4×. Scale bars: 500 nm (upper panel), 200 nm (lower panels).

D   Line scans through Bassoon staining (green) in pre-synaptic active zones and through Homer 1 staining (magenta) in the corresponding post-synaptic densities reveal the distance between the two. The image inset shows three example line scans and identifies them by number.

E   Line scans through RIM1/2 staining (green) in pre-synaptic active zones and through PSD95 staining (magenta) in the corresponding post-synaptic densities reveal the distance between the two. The image inset shows three example line scans and identifies them by number.

F   Histogram showing the distribution of Bassoon to Homer 1 distances and RIM1/2 to PSD95 distances (*n* = 15 neuronal areas, with the corresponding synapses, for Bassoon and Homer 1, *n* = 74 neuronal areas for RIM1/2 and PSD95).

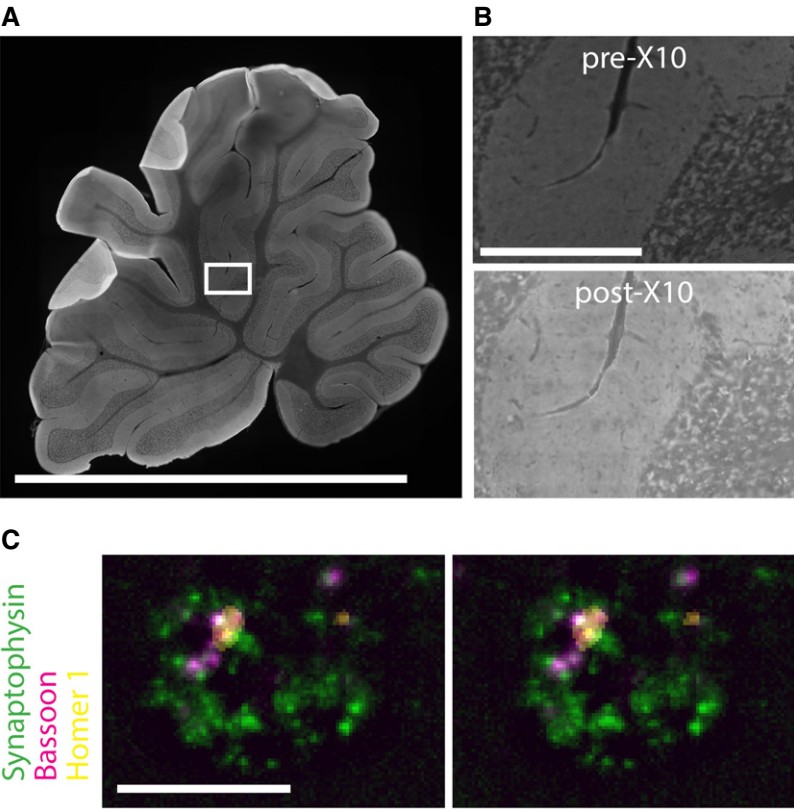

**Figure 5.  Multi-color X10 imaging in brain slices.**

A  Overview image of a rat cerebellum brain slice before expansion showing Synaptophysin staining. Scale bar: 0.5 cm. Note that the image is stitched together from multiple imaging frames.

B  Magnification of the section framed in (A), before expansion (top panel) and after expansion (bottom panel). Expansion factor: 9.6×. Scale bar: 250 μm. Note that both images are stitched together from multiple imaging frames.

C  A stereo view of a single synapse in the brain slice (from the boxed region in panel A). Individual synaptic vesicles (identified by Synaptophysin, green), along with the active zone (identified by Bassoon, red) and the post-synaptic density (identified by Homer 1, yellow) are evident. Scale bar: 1 μm.

that of X10 on epifluorescence microscopes. However, combining X10 with physics-based super-resolution, and especially with a coordinate-targeted approach such as STED [32], which can be applied rapidly and efficiently to large imaging volumes, would provide an ultimate resolution equal to the size of the fluorophores. This should enable the investigation of molecular assemblies or molecule orientation more efficiently than virtually any other current tools, especially if probes smaller than antibodies are employed [13,14,21].

## Materials and Methods

### Cell culture

Neuronal hippocampal cultures were obtained from dissociated hippocampi of newborn Wistar rats of mixed gender as described elsewhere [33,34]. In brief, brains were extracted from the skulls of 2-day-old rat pups, and the hippocampi were isolated under a dissection microscope. Following washes with HBSS (Invitrogen, Waltham, MA, USA) to remove tissue debris, the hippocampi were incubated for 1 h in enzyme solution (10 ml DMEM, 2 mg

cysteine, 100 mM $CaCl_2$, 50 mM EDTA, and 25 U papain), equilibrated with carbogen for 10 min, and sterile-filtered. Before mechanical dissociation, cells were washed thoroughly with HBSS and were incubated for 15 min in inactivating solution (2 mg albumin and 2 mg trypsin inhibitor in 10 ml FCS-containing DMEM). Before seeding, coverslips were treated with nitric acid, washed thoroughly with $ddH_2O$, sterilized, and coated overnight with 1 mg/ml PLL. After coating, coverslips were washed thoroughly with sterile $ddH_2O$ and were incubated with plating medium (DMEM supplemented with 10% horse serum, 3.3 mM glucose, and 2 mM glutamine). Following dissection, neurons were plated at a concentration of ~30,000/cm² and were left to adhere for 1–4 h at 37°C in a 5% $CO_2$ cell incubator. After adhesion, the medium was changed to Neurobasal-A medium containing 1:50 B27 supplement and 1:100 GlutaMAX (all Gibco, Life Technologies, Carlsbad, CA, USA). Neurons were kept in culture at 37°C and 5% $CO_2$ for 14–21 days before use.

Certified mycoplasma-free COS7 cells were originally sourced and authenticated from the German Collection of Microorganisms and Cell Cultures (Leibniz-Institute DSMZ, Braunschweig, Germany). COS7 cells were cultured in DMEM with 4 mM

L-glutamine and 10% FCS (Thermo Fisher Scientific, Waltham, MA, USA), supplemented with 60 U/ml of penicillin and 0.06 mg/ml streptomycin (Sigma-Aldrich, St. Louis, MO, USA) at 37°C and 5% $CO_2$ according to standard protocols, and were used 2–3 days after passage. For tubulin stainings, the cells were plated on 18-mm coverslips coated with 0.1 mg/ml PLL for 1 h.

## Brain slices

Brain slices were prepared from adult (6–8 weeks old) Wistar rats euthanized with $CO_2$, by perfusion with PBS to remove blood, followed by perfusion with 4% PFA in PBS for fixation. The brains were then removed from the skull and were placed into 4% PFA in PBS at 4°C overnight. The brains were then transferred to a solution of 30% sucrose in PBS on 4°C until they sank to the bottom of the solution, before freezing and storing them on −80°C until use. The brains were cut into 20- to 30-μm-thick slices on a Leica CM1850 cryotome (Leica, Wetzlar, Germany).

## Production and labeling of single-chain variable fragments (scFv)

Single-chain variable fragments (scFv) used for tubulin stainings (Appendix Fig S1) were previously described [35] and were produced in the laboratory as fusion proteins with scSUMO and a 14×-His-tag at the N-terminus, and twin Strep-tag at the C-terminus. These constructs were expressed in shuffle express bacteria cultured in TB medium (Sigma-Aldrich, St. Louis, MO, USA) at 25°C overnight. After harvesting the bacteria by centrifugation, the pellet was resuspended in His-tag binding buffer (50 mM HEPES, 150 mM NaCl, 4.5 mM $MgCl_2$, pH 8.0). Lysis of bacteria was achieved by sonication, and debris was removed by centrifugation for 1 h at 16,000 g. The supernatant was then filtered through a 0.45-μm syringe filter (Sartorius, Göttingen, Germany). The scFv were then immobilized on Ni-beads (cOmplete resin; Roche, Basel, Switzerland) for 1 h before 3× washing with high-salt His-tag buffer (His-tag buffer + 350 mM NaCl). The scFv were then eluted by cutting with SUMO protease (~30 min). All scFv isolation steps were performed at 4°C. The scFv were then further purified (buffers for washing and elution as above, but with 1 mM EDTA) via the twin Strep-tag with an ÄKTA machine (GE Healthcare, Little Chalfont, UK). The eluted fraction was then concentrated using Amicon cutoff filters (Sartorius, Göttingen, Germany).

The scFv were conjugated to maleimide-Atto647N (Abberior, Göttingen, Germany). For this, 20–30 nmol of scFv was solved in 500 μl amine-free buffer (i.e. HEPES-buffered saline solution) + 50 μl 1 M $NaHCO_3$, and the pH was adjusted to 8.6. The NHS-reactive dye (see above) was then added for 1 h in the dark at more than 4× molar excess over the scFv. To quench unreacted dye, 50 μl 1 M Tris (pH 8.0) was then added, and the labeled scFv were purified via Sephadex columns (G25, superfine; GE Healthcare, Little Chalfont, UK). Eluted fractions were then collected and, if necessary, pooled and stored in 50% glycerol at −20°C.

## Immunostainings

Immunostainings of cultured rat hippocampal neurons were performed after fixation in 4% PFA, for 10 min on ice followed by 30 min at room temperature, and quenching with 100 mM $NH_4Cl$ for 20 min. The neurons were then blocked and permeabilized in PBS (137 mM NaCl, 2.7 mM KCl, 10 mM $Na_2KPO_4$, 2 mM $KH_2PO_4$, pH 7.3) + 2.5% BSA + 0.1% Triton X-100 3 × 5 min. Immunostaining was carried out in the same solution, with added primary antibodies, by placing the coverslips upside down on 80 μl of staining solution on Parafilm in a humidified chamber, for 1 h. The samples were then washed 3 × 5 min in the same solution without antibody, before secondary antibody incubation, which was performed in the same fashion as primary antibody incubation. The samples were then washed 3 × 5 min in PBS + 2.5% BSA, 3 × 5 min in high-salt PBS (PBS + 350 mM NaCl), and 2 × 5 min in PBS.

COS7 cells were fixed in methanol at −20°C (for 20 min) for tubulin stainings. Immunostainings were then performed as described above for the neurons, but without Triton X-100.

Brain slices were subjected to antigen retrieval in 10 mM citrate (Sigma-Aldrich, St. Louis, MO, USA) + 0.5% Tween-20 (Merck, Darmstadt, Germany) at 80°C before staining. The brain slices were then blocked and permeabilized in PBS + 2.5% BSA + 0.1% Triton X-100, three times for 10 min, before incubation with primary antibodies in the same solution, at 4°C overnight. The brain slices were then washed in the same solution without antibody 3 × 10 min before incubation with secondary antibodies in the same solution, for 2–3 h at room temperature. The brain slices were then washed 3 × 10 min in PBS + 2.5% BSA, 3 × 10 min in high-salt PBS, and 2 × 10 min in PBS.

The following primary antibodies were used: rat monoclonal anti-tubulin (MA1-80017; Thermo Fisher Scientific, Waltham, MA, USA), anti-tubulin single-chain variable fragments (scFv) directly conjugated to Alexa Fluor 488 for X10 or Atto647N for STED imaging (self-produced, see above), guinea pig polyclonal anti-Synaptophysin (101 004; Synaptic Systems, Göttingen, Germany), rabbit polyclonal anti-Homer1 (160 003; Synaptic Systems), mouse monoclonal anti-Bassoon (SAP7F407; Enzo, Farmingdale, NY, USA), rabbit polyclonal anti-VDAC (sc-98708; Santa Cruz, Heidelberg, Germany), mouse monoclonal anti-PSD95 (MA1-046; Thermo Fisher Scientific, Waltham, MA, USA), and rabbit anti-RIM1 (140 003; Synaptic Systems). The following secondary antibodies were used: donkey anti-rat conjugated to Alexa Fluor 488 (712-545-153; Dianova, Hamburg, Germany), goat anti-rabbit conjugated to Alexa Fluor 546 (A-11035; Thermo Fisher Scientific), goat anti-rat conjugated to CF647 (#20013; Biotium, Fremont, CA, USA), and donkey anti-mouse conjugated to CF633 (#20124; Biotium, Fremont, CA, USA). All antibodies were used at a dilution of 1:100 from 0.4 to 1 mg/ml stocks.

## Anchoring (protein retention)

To prepare antibodies for linkage into the gel, samples were incubated overnight at room temperature in PBS + 0.1 mg/ml Acryloyl-X (Life Technologies, Carlsbad, CA, USA), as described before [9] (the anchoring time can be shortened, but should be at least 6 h). We found that signal retention is highest and that distortions during expansion are lowest, when gelation (incubation with the gel solution) is performed immediately after Acryloyl-X treatment. Anchoring of tissue samples can be optimized by using 150 mM $NaHCO_3$ instead of PBS.

## X10 microscopy gel preparation

The gel recipe was adapted from Cipriano *et al* [11] and is prepared as follows. The gelling solution is prepared by dissolving 33% (w/w) of *N,N*-dimethylacrylamide acid (DMAA) and sodium acrylate (SA) monomers at a molar ratio of 4:1 (DMAA:SA) in $ddH_2O$. To illustrate, for 10 ml of solution this would mean 6.7 g $ddH_2O$, 2.67 g DMAA, and 0.64 g SA. It is advised to use SA of high purity, which can be checked by dissolving it in $ddH_2O$ at a concentration of 0.38 g/ml; if the solution is mostly clear and without yellow tint, the SA can be used. The solution was then bubbled for 40 min at room temperature with $N_2$ to remove molecular oxygen from the solution, because oxygen inhibits the polymerization reaction. Immediately afterward, the initiator potassium persulfate (KPS) was added at 0.4 molar% relative to the monomer concentration (0.036 g in case of 10 ml of solution) and the solution was bubbled with $N_2$ for an additional 15 min, on ice (to inhibit premature gelation). Precision can be improved by preparing a KPS stock of 0.036 g/ml, reducing the amount of $ddH_2O$ in the monomer solution by 10% of the total volume (so 5.7 g instead of 6.7 g for 10 ml of monomer solution in the example described above), and adding from the KPS stock at a ratio of 1+9 (so 1 ml to bring the volume of 9 ml to a total of 10 ml in the example described above). The KPS stock should be prepared fresh every time, as stability of KPS in an aqueous stock is poor. Subsequently, 40 μl of *N,N,N′,N′*-tetramethyl-ethane-1,2-diamine (TEMED) was added on 10 ml gelation solution to accelerate the polymerization reaction and the sample was placed into the gelation solution and was incubated in a humidified chamber at room temperature for 6–24 h. The progress of gelation can be tested by including gels without sample, i.e. gels on empty coverslips, and removing them from the gelation chamber; gelation is complete when the gel detached from the coverslip smoothly and without residue. For cells cultured on a coverslip, this was done by removing all excess buffer with a tissue and then placing the coverslip upside down on an 80 μl droplet of gel solution on Parafilm; alternatively, a custom gelation chamber can be constructed as described previously [7]. For brain slices, we performed pre-incubation with the monomer solution, two times for 20 min each, at room temperature, followed by 10 min with the monomer solution and added KPS on ice, before placing the brain slices in a custom gelation chamber. For brain slices, twice the usual amount of TEMED was added (80 μl for 10 ml gel solution) to the monomer solution, to account for dilution of the final gel solution through carry-over of liquid volume when moving the brain slices with a brush from the pre-incubation solution to the final gelation chamber. All chemicals used were from Sigma-Aldrich, St. Louis, MO, USA.

## X10 gel digestion and expansion

The polymerized gels were removed from Parafilm or from the gelation chambers after polymerization had been achieved, and were placed into digestion buffer (50 mM Tris buffer, 0.8 M guanidinium chloride, 8 U/ml proteinase K, 0.5% Triton X-100, pH 8.0). All chemicals used were from Sigma-Aldrich, St. Louis, MO, USA. Digestion was carried out at 50°C in a humidified chamber overnight. The gels were placed directly into pre-warmed digestion buffer. Shorter digestion times at room temperature can lead to tears in the sample (Appendix Fig S1) during expansion, due to the

increased expansion factor and due to the tendency of the gel described here to moderately (~1.5–2×) expand during digestion. Digestion time should be at least 12 h. The digested samples were then placed into an excess volume of $ddH_2O$ (at least 10-times the final gel volume) for expansion. The $ddH_2O$ was replaced 5–6 times, with 20–30 min per expansion step, to reach the final expansion of ~10×. The maximized expansion, up to ~11.5×, can often be achieved by increasing the incubation times with $ddH_2O$ during each step to at least 1 h, increasing the number of buffer exchanges, and including a final overnight expansion step.

## Spleen tissue expansion

We obtained spleen tissue slices from BioCat (Heidelberg, Germany). These were fixed with 4% PFA for 15 min and were expanded in a similar fashion to brain tissue slices, with the following exceptions. First, the anchoring reaction was performed with 0.2 mg/ml Acryloyl-X (NHS ester), in bicarbonate buffer (150 mM $NaHCO_3$, pH 8.3). Second, the proteinase K digestion was performed at 50°C, overnight, in the same buffer indicated above, but with the addition of 2 mM $CaCl_2$. For simplicity, and to visualize all protein components of the tissue, these samples were labeled with Alexa Fluor 488 NHS ester, by incubation in bicarbonate buffer for at least 2 h (using 1 mg/ml concentrations of the dye).

## Imaging

All expansion microscopy imaging was performed on a Nikon Ti-E epifluorescence microscope (Nikon Corporation, Chiyoda, Tokyo, Japan) equipped with a 10× 0.45 NA air Plan Apochromat objective, a 20× 0.75 NA air Plan Apochromat objective, a 60× 1.40 NA oil-immersion Plan Apochromat objective, a 100× 1.4 NA oil-immersion Plan Apochromat objective (all from Nikon Corporation, Chiyoda, Tokyo, Japan), and a 150× 1.45 NA oil-immersion Plan Apochromat objective (Olympus, Shinjuku, Tokyo, Japan), with an HBO-100W lamp, an IXON X3897 Andor camera (Andor, Belfast, Northern Ireland, UK), or a Nikon DS-Qi2 camera (Nikon Corporation, Chiyoda, Tokyo, Japan), and operated with the NIS-Elements AR software (Nikon Corporation, Chiyoda, Tokyo, Japan). The camera gain and exposure time were adjusted for each sample individually to compensate for the loss of fluorescence during expansion, to achieve a comparable level of intensity in the images collected before and after expansion. STORM imaging was performed on a setup based on an Olympus IX83 body (Olympus, Shinjuku, Tokyo, Japan), equipped with a 100x Plan Apochromat objective (Olympus, Shinjuku, Tokyo, Japan), an iCHROME MLE laser source (Toptica, Gräfeling, Germany), and an Andor iXON Ultra 888 EMCCD camera (Andor, Belfast, Northern Ireland, UK), and operated with Olympus cellSens software (Olympus, Shinjuku, Tokyo, Japan). For STORM imaging, blinking was induced with the following buffer: 50 mM Tris, 10 mM NaCl, 10 mM MEA, 10% glucose, 2,000 U/ml catalase, 50 U/ml glucose oxidase. STORM images were analyzed with the rapid*STORM* software [36]. STED imaging was performed on a setup based on an Olympus IX83 body (Olympus, Shinjuku, Tokyo, Japan) equipped with a 100× 1.40 NA oil-immersion Plan Apochromat objective (Olympus, Shinjuku, Tokyo, Japan), a 19″ pulsed 640-nm excitation laser, and a 775-nm depletion laser, and operated with Imspector software (all

Abberior Instruments GmbH, Göttingen, Germany). Alternatively, for the images shown in Appendix Fig S1C and D, STED imaging was performed on a TCS SP5 microscope (Leica, Wetzlar, Germany) equipped with a HCX 100× 1.4 NA oil-immersion Plan Apochromat STED objective, a pulsed diode excitation laser at 640 nm (PDL 800-D; PicoQuant, Berlin, Germany), and a Mai Tai Ti:Sapphire depletion laser at 750 nm (Spectra-Physics, Mountain View, CA, USA), and operated with the LAS AF imaging software (version 2.7.3.9723; Leica, Wetzlar, Germany).

### Image analysis and data evaluation

The imaging data were analyzed using custom-written MATLAB (MathWorks, Natick, MA, USA) routines, ImageJ (Wayne Rasband, NIH, Bethesda, MD, USA), and SigmaPlot (Systat Software Inc., Erkrath, Germany). For presentation purposes, the image in Fig 3A, bottom right, was deconvolved using Huygens Essential 4.4 (Scientific Volume Imaging, Hilversum, Netherlands), based on inbuilt algorithms that were adjusted to the imaging parameters of the particular image. The same was performed for several supplementary movies, as indicated.

### Determination of expansion factors

The expansion factor for each experiment was determined by direct comparison of pre-expansion and post-expansion images. For this, we measured the physical distances between landmark positions in the pre-expansion images, then we measured the physical distances of the same landmark positions in the corresponding post-expansion images, and finally we determined the expansion factor by simple division of the latter over the former. We performed at least 10 such distance comparisons for each experiment and used the average as expansion factor, which is listed for each individual figure panel in the corresponding figure legend of this manuscript.

### Determination of resolution

The resolution was determined by measuring the full width at half maximum (FWHM) of the edges of Pmp70-stained peroxisomes, in the raw post-expansion images (no deconvolution), and averaging the result (Fig 3A). We then compared these data to simulated peroxisomes, where we used realistic antibody lengths and random orientation of primary/secondary antibody complexes to determine what the maximum theoretical resolution would be that we could achieve in this model system. As the resolution (FWHM) determined in this way matched very closely between model (22.8 nm) and experiment (25.2 nm), we used the parameters of the model to also simulate expanded microtubules. We confirmed that an expansion factor of 11.5×, the maximum we could achieve here, is just below the limit we would require to resolve the microtubule lumen. The simulations were performed using custom-written MATLAB routines as follows. For peroxisomes, spheres with a diameter of 100 nm were simulated and were decorated with primary and secondary antibody complexes. The complexes were assumed to be rigid and randomly oriented, thus placing the fluorophores at variable distances from the epitopes (between 0 and 25 nm, the maximum length of a primary/secondary antibody complex). The locations of the fluorophores were overlapped with experimentally measured PSFs for the different microscopes used (epifluorescence, STED, or STORM), and 2D images were thus derived, taking into account the fact that large expansion factors place some of the fluorophores out of the focus volume. The antibody orientations were unconstrained, as the relative orientations of the Pmp70 epitopes are probably randomized after fixation and permeabilization. The appropriate expansion factor was used in determining the positions of the fluorophores. A similar approach was used for microtubules, but with more restrictively positioned antibodies, accounting for the fact that they cannot penetrate inside the microtubules.

### Determination of distortions

To estimate the distortion factor during expansion, corresponding fixed positions were marked in pre-expansion images and post-expansion images, and their relative shift in position was evaluated. For this, the pre- and post-expansion images were first aligned on a shared central recognizable landmark in the images and were then manually rotated to achieve the best fit. This was followed by an automated analysis, described in the next phrases, which accounted for any errors caused by imprecise manual alignments, and which was corrected for distortions induced by the manual handling of the pre-expansion samples during imaging. After having aligned the pre- and post-expansion images in their central landmark position, we determined the shift between the features in the pre- and post-expansion images as a function of distance from the aligned central landmark. To determine this shift, we generated boxes of 400 by 400 pixels in both images and determined their correlation. Then, the box from the post-expansion image was shifted in different directions, until the best fit was obtained. The boxes were first placed in the central landmark position and were then moved pixel by pixel toward the image periphery. This provides a measure of the distortion-induced shift between the two images, as a function of the distance from the perfectly aligned image center. This method was chosen as it was the only automatic analysis that we could easily apply to these data. The differences between the pre-expansion epifluorescence image and the post-expansion super-resolution image are too large for accurate implementation using methods such as the B-spline, as used before in evaluating 4× expansion microscopy images.

**Expanded View** for this article is available online.

## Acknowledgements

We thank Martin Helm and Felipe Opazo for help with the STED imaging. We thank Johann G. Danzl for help with the confocal imaging. S.T. was supported by an Excellence Stipend of the Göttingen Graduate School for Neurosciences, Biophysics, and Molecular Biosciences (GGNB). This work was supported by grants to S.O.R. from the European Research Council (ERC-2013-CoG NeuroMolAnatomy) and from the Deutsche Forschungsgemeinschaft (DFG): SFB 889/A5, 1967/7-1, SFB1190/P09, and SFB1286/Z03.

## Author contributions

The manuscript was written through contributions of all authors. All authors have given approval to the final version of the manuscript. ST and DC performed the stainings. MM cultured COS7 cells, produced the labeled scFv for tubulin stainings, and performed the STED imaging. HW performed the

STORM imaging. ST performed all other experiments and the expansion microscopy imaging. ST, SK, and SOR conceived the project. ST and SOR performed the data evaluation and wrote the manuscript.

## Conflict of interest

The authors declare that they have no conflict of interest.

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
