## [Review Process File · EMBO Reports]

X10 Expansion Microscopy Enables 25 nm Resolution on Conventional Microscopes

Sven Truckenbrodt, Manuel Maidorn, Dagmar Crzan, Hanna Wildhagen, Selda Kabatas, Silvio O. Rizzoli

Review timeline:

Submission date:	20 January 2018
Editorial Decision:	16 February 2018
Revision received:	11 May 2018
Editorial Decision:	4 June 2018
Revision received:	8 June 2018
Accepted:	18 June 2018

Editor: Esther Schnapp/Martina Rembold

Transaction Report:

1st Editorial Decision

16 February 2018

Thank you for the submission of your research manuscript to our journal. We have now received the full set of referee reports that is copied below. Since Esther Schnapp is currently traveling, I have temporarily taken over the handling of your manuscript.

As you will see, the referees acknowledge the potential interest of the X10 method. However, referee 2 and 3 also make a number of suggestions on how the manuscript could be improved to show the full potential of the method but also to analyse and discuss potential limitations of it, and I think that all of them should be addressed.

Given these constructive comments, we would like to invite you to revise your manuscript with the understanding that the referee concerns must be fully addressed and their suggestions taken on board. Please address all referee concerns in a complete point-by-point response. Acceptance of the manuscript will depend on a positive outcome of a second round of review. It is EMBO reports policy to allow a single round of revision only and acceptance or rejection of the manuscript will therefore depend on the completeness of your responses included in the next, final version of the manuscript.

Revised manuscripts should be submitted within three months of a request for revision; they will otherwise be treated as new submissions. Please contact us if a 3-months time frame is not sufficient for the revisions so that we can discuss the revisions further.

Supplementary/additional data: The Expanded View format, which will be displayed in the main HTML of the paper in a collapsible format, has replaced the Supplementary information. You can submit up to 5 images as Expanded View. Please follow the nomenclature Figure EV1, Figure EV2 etc. The figure legend for these should be included in the main manuscript document file in a section called Expanded View Figure Legends after the main Figure Legends section. Additional Supplementary material should be supplied as a single pdf labeled Appendix. The Appendix

includes a table of content on the first page including page numbers, all figures and their legends. Please follow the nomenclature Appendix Figure Sx throughout the text and also label the figures according to this nomenclature. For more details please refer to our guide to authors.

Regarding data quantification, please ensure to specify the name of the statistical test used to generate error bars and P values, the number (n) of independent experiments underlying each data point (not replicate measures of one sample), and the test used to calculate p-values in each figure legend. Discussion of statistical methodology can be reported in the materials and methods section, but figure legends should contain a basic description of n, P and the test applied.

We now strongly encourage the publication of original source data with the aim of making primary data more accessible and transparent to the reader. The source data will be published in a separate source data file online along with the accepted manuscript and will be linked to the relevant figure. If you would like to use this opportunity, please submit the source data (for example scans of entire gels or blots, data points of graphs in an excel sheet, additional images, etc.) of your key experiments together with the revised manuscript. Please include size markers for scans of entire gels, label the scans with figure and panel number, and send one PDF file per figure.

- a complete author checklist, which you can download from our author guidelines (<http://embor.embopress.org/authorguide#revision>). Please insert page numbers in the checklist to indicate where the requested information can be found.
 - a letter detailing your responses to the referee comments in Word format (.doc)
 - a Microsoft Word file (.doc) of the revised manuscript text
 - editable TIFF or EPS-formatted figure files in high resolution
- (In order to avoid delays later in the publication process please check our figure guidelines before preparing the figures for your manuscript:
http://www.embopress.org/sites/default/files/EMBOPress_Figure_Guidelines_061115.pdf)
- a separate PDF file of any Supplementary information (in its final format)
 - all corresponding authors are required to provide an ORCID ID for their name. Please find instructions on how to link your ORCID ID to your account in our manuscript tracking system in our Author guidelines (<http://embor.embopress.org/authorguide>).

As part of the EMBO publication's Transparent Editorial Process, EMBO reports publishes online a Review Process File to accompany accepted manuscripts. This File will be published in conjunction with your paper and will include the referee reports, your point-by-point response and all pertinent correspondence relating to the manuscript.

I look forward to seeing a revised version of your manuscript when it is ready. Please let me know if you have questions or comments regarding the revision.

REFeree REPORTS

Referee #1:

I have reviewed an earlier version of this manuscript for before and was supportive of publication and I am now. Expansion microscopy is a method that has been first described 2.5 years ago and

development is moving quickly to realise its enormous promise. The contribution by Rizzoli et al is likely to be important in the further development of the field and should be published without further delay.

Here's my previous review:

My concerns have been addressed with convincing new experimental data and theoretical analysis. The data are of at least similar quality to that of the recently published iterative expansion microscopy and achieving 10 x in a single step rather than in two steps seems less error prone. I believe this substantial advance on an extremely timely and important problem should be published without further delay. Like the original expansion microscopy this new approach using a well-suited new polymer now needs to be tested in the community and the manuscript gives detailed explanation on how to proceed with this. The work is a key enabling step towards finding the optimal technical approach to realise the full potential of the conceptual breakthrough of expansion microscopy.

Referee #2:

Here, Truckenbrodt et al., presents a super resolution microscopy approach based on tissue expansion. The authors developed "one embedding and one expansion step" reaching down to 20-30 nm resolution, which can be used in multiple color channels without any complicated procedure. Authors also note that the method works only with fluorescent dyes but the fluorescent proteins.

Overall, the authors present convincing data, which could be improved to show full potential of the method. Therefore, I would recommend publication of the work if authors could address the following points:

- By 2nd Ab staining, what we see is the chemical dye attached to Ab, which has much smaller size than antibody itself. Therefore, I am not really sure how much the argument on antibody size limits the resolution is valid. Do the authors know exactly how many fluorescent dye molecules are attached to one secondary antibody? Can a labeling method that does not require antibodies be used to test the method- for example tracer/dyes for actin, microtubules or organelles? Could Alexa dyes be added on to highly phagocytotic cells and be imaged by X10 to resolve dyes as much as possible?
- While the authors claim that X10 also works with tissues, I did not find Fig 5 as convincing as in vitro results. Please provide quantifications and at least one more example from another tissue labeling to substantiate this claim.
- I am not clear exactly why the authors did not present confocal images of X10 samples. If the aim is to use a conventional microscope, that is ok. But still, a confocal stack would be very useful especially considering the presented 3D movies, which are mostly out of focus.
- Please add scale bars to all videos.

Referee #3:

Expansion microscopy achieves super-resolution capabilities by physical expansion of the tissue specimen and using conventional confocal or epifluorescence microscopes rather than sophisticated STORM or STED set-ups. Expansion microscopy techniques are easy to implement across Labs and have wide-spread biological applications as they can resolve closely apposed proteins and cell-compartments. In this EMBO report, Truckenbrodt et al describe a technique for a 10 fold expansion microscopy which is a 2.5 fold increment of the currently available 4 fold expansion microscopy procedures. This technique of x10 microscopy allows the authors to achieve a 25nm resolution, which is normally achievable only by STED or STORM techniques or by using the expansion microscopy procedure iteratively. The paper comes out of the Rizzoli Lab which is well-renowned in using sophisticated microscopy approaches when asking questions in neuroscience. The x10 method described in this report would be of interest to a wide-range of molecular and cellular biologists.

I have the following concerns:

1. There are caveats to this x10 approach as compared to the conventional 4x expansion microscopy method the most striking being that intrinsic fluorescent protein labels routinely being used in transgenic approaches, and which remains conserved in the 4x expansion procedure is lost in this x10 approach. Further, considering the 10-fold separation between cellular compartments such as between pre- and postsynaptic compartments, one wonders if translinking protein interactions that normally span across the synapse are preserved with this x10 approach. Have the authors any data on whether labeling for a transsynaptic protein is possible with this x10 approach? The authors could either test for this if they can label for a transsynaptic protein with their method or they should speculate about this limitation to the applicability of their method in the discussion.

2. In Figure 4A the reader cannot see individual synaptic vesicles with the synaptophysin labeling- each synaptic vesicle should be about 40nm and thus well resolved by the authors x10 approach which allows a resolution of 25nm. Part of the problem might be that the images are taken with an epifluorescent scope and thus capturing the out of focus fluorescence. The authors should re-acquire the images with a conventional confocal microscope so the readers can better resolve the data in Figure 4 and also for the brain slices images shown in Figure 5. This should be easily addressable and is important to enhance the quality of data in Figure 4, Figure 5 and the associated suppl figures. If labeling with synaptophysin cannot allow clean punctate label then the authors should consider including data for presynaptic proteins that have more punctate expression such as vGlut.

3. The images of Bassoon and Homer1 from hippocampal neurons in Figure 4 should include a pre-expansion image of the two synaptic markers beside the x10 expanded images. This will enable the reader to directly relate the effectiveness of the approach. A pre-expansion image of these markers is shown in Suppl Figure 6 but it is difficult to gather which field of view the supplemental image has been taken from. The main figure 4 itself should show pre- and post-expansion images of these synaptic markers for the same region/field of view.

4. Figure 5C: It is unclear to the reader which part/region of the brain slice has been selected to show the three synaptic markers: please include an image of the three synaptic markers across the entire brain slice and place a box or a circle indicating the area represented in panel C.

5. Supplemental Figure 7: It is difficult for the reader to discern the synaptic markers in panel C and the associated movie - please remake the image stack with a confocal scope so the diffuse (green) signal can be better resolved.

6. Supplemental Figure 7: No Homer signal is evident in panel B although the label suggests that there should be.

7. While referring to the images from rat cerebellum the authors refer to 'the directional orientation of synapses'. It is unclear what is meant by this phrase and the authors should either clarify what they mean or exclude the use of such jargon.

8. Similar concern for the phrase 'morphology of these structures' used in page 9 of the text. It is not directly apparent to the reader what is meant. Please clarify or omit.

1st Revision - authors' response

11 May 2018

Referee #1:

I have reviewed an earlier version of this manuscript for before and was supportive of publication and I am now. Expansion microscopy is a method that has been first described 2.5 years ago and development is moving quickly to realise its enormous promise. The contribution by Rizzoli et al is likely to be important in the further development of the field and should be published without further

delay.

Here's my previous review:

My concerns have been addressed with convincing new experimental data and theoretical analysis. The data are of at least similar quality to that of the recently published iterative expansion microscopy and achieving 10 x in a single step rather than in two steps seems less error prone. I believe this substantial advance on an extremely timely and important problem should be published without further delay. Like the original expansion microscopy this new approach using a well-suited new polymer now needs to be tested in the community and the manuscript gives detailed explanation on how to proceed with this. The work is a key enabling step towards finding the optimal technical approach to realise the full potential of the conceptual breakthrough of expansion microscopy.

We thank the referee for the comments.

Referee #2:

Here, Truckenbrodt et al., presents a super resolution microscopy approach based on tissue expansion. The authors developed "one embedding and one expansion step" reaching down to 20-30 nm resolution, which can be used in multiple color channels without any complicated procedure. Authors also note that the method works only with fluorescent dyes but the fluorescent proteins.

Overall, the authors present convincing data, which could be improved to show full potential of the method.

We thank the referee for the comments.

Therefore, I would recommend publication of the work if authors could address the following points:

- By 2nd Ab staining, what we see is the chemical dye attached to Ab, which has much smaller size than antibody itself. Therefore, I am not really sure how much the argument on antibody size limits the resolution is valid. Do the authors know exactly how many fluorescent dye molecules are attached to one secondary antibody? Can a labeling method that does not require antibodies be used to test the method- for example tracer/dyes for actin, microtubules or organelles? Could Alexa dyes be added on to highly phagocytotic cells and be imaged by X10 to resolve dyes as much as possible?

The point of the referee is a very important one. Indeed, one detects the fluorescence dyes, not the antibodies, and in principle every fluorescent dye molecule will provide a spot which is only limited by diffraction, not by the size of the antibody.

However, this consideration only applies to a situation in which a single dye is linked to a single target. In most immunostainings we are faced with a situation in which several target proteins are grouped in a cluster or in an organelle, and each is labeled by a primary antibody, which is in turn revealed by fluorophore-conjugated secondary antibodies. The antibodies place the fluorophores at a considerable displacement from the target proteins. This results in a cloud of fluorescence around the original position of the target proteins. The size of this cloud limits the level of detail that can be perceived in the resulting image. A particularly clear example was shown for microtubules imaged using antibodies (~15 nm in length) or nanobodies (~2-4 nm in length) by Mikhaylova *et al.*, Nature Communications, 2015 (figures 1 and 2). The nanobodies indicated a microtubule diameter size of ~20 nm smaller than that measured with antibodies (figure 2 of the respective publication).

It was difficult to perform this type of test in expansion microscopy. Small labels, including SNAP tags or nanobodies, are impossible to use, as they are lost during the enzymatic homogenization of the samples. This has been also the experience of other investigators (Gao *et al.* (2018) Expansion stimulated emission depletion microscopy (ExSTED). bioRxiv, doi: <https://doi.org/10.1101/278937>). We are currently developing nanobodies adapted for expansion microscopy, but this is beyond the purpose of the current manuscript.

However, to address the point of the referee directly, we performed the following experiment. In separate experiments, we applied single fluorophore-conjugated antibodies on coverslips, or applied

single primary antibodies to similar coverslips, which were then immunostained by fluorophore-conjugated secondary antibodies. We then expanded the samples, and imaged them using an epifluorescence microscope, using a highly sensitive camera.

As indicated in the new Appendix Fig. S3, the single antibodies provided a full width at half maximum (FWHM) of ~25 nm, while the primary antibodies stained by secondary antibodies formed spots with an average FWHM of 32.9 nm, which demonstrates that the antibody clusters indeed limit the perceived resolution under such expansion conditions.

We would also like to point out that this is an extreme example of antibody-induced disturbance of resolution. Here the antibodies are probably oriented in all directions, randomly, and provide a large blurred spot. In many real immunostainings the antibodies may be oriented in specific directions (*e.g.* perpendicular to the membrane of an organelle), in which case their influence on resolution may be lower. Also, the primary antibodies on coverslips are bound by three secondary antibodies, on average. This value may be lower in real samples, which again may reduce the perturbing effects.

Finally, we did attempt to analyze a non-specific staining, as suggested by the referee, using fluorescent dyes or a membrane-binding molecule (Wheat Germ Agglutinin conjugated to Alexa488), but the very variable sample geometry prohibited serious resolution analyses, which were far easier in the simpler antibodies-on-coverslips experiments.

We have also adjusted the paragraph that the referee referred to, stating that the large size of the antibodies “effectively limits the level of detail that can be observed”, rather than “the resolution”, which should make our meaning clearer to the readers.

- While the authors claim that X10 also works with tissues, I did not find Fig 5 as convincing as in vitro results. Please provide quantifications and at least one more example from another tissue labeling to substantiate this claim.

The referee points here to a very important issue, which we have decided to address thoroughly.

In brief, X10 does work with tissues, but expansion procedures are more easily applied to cell cultures than to tissues. The community is often experiencing difficulties with tissue expansions, and many protocols are currently circulated, attempting to adapt the technology to different types of tissues (*e.g.* multiple articles from the Boyden laboratory, targeted to differently embedded tissues).

Our laboratory has specialized in neuronal and brain samples for more than a decade, and we have therefore optimized X10 for brain slices. We now include a new Appendix Fig. S5, in which we quantified a number of parameters for the brain slices:

- We analyzed the distortions induced by X10 expansion in brain slices, by comparing pre- and post-expansion images. The distortions are comparable with those seen in cultures.
- We analyzed the distance between the pre- and postsynaptic proteins Bassoon and Homer, and we found this to be very similar to that measured in neuronal cultures for the same proteins.
- We analyzed the number of synaptic vesicles per synapse, and their diameter. Both parameters fit very well with the previous knowledge from electron microscopy studies of the same types of brain sections.

To address the comment of the referee more directly, we have also tested the performance of X10 on a different and more difficult type of sample (see Appendix Fig. S6). Several companies provide thin-sectioned tissues that have been snap-frozen, and have been mounted on positively-charged slides. Such tissues are partially dried, due to the procedures involved with their shipping and storage. In addition, penetration of proteases is here far more difficult than in free-floating fresh samples (as for the brain sections), thus providing a more stringent test to any homogenization and expansion procedure.

We purchased spleen tissue sections from BioCat (www.biocat.com), and we adapted X10 for their analysis. Using the same protocol as for cell cultures and brain sections proved futile: the sections disintegrated, and no expansion could be obtained. This was due to the fact that these tissues required more thorough homogenization and gel anchoring. We achieved this by:

- 1) Performing the gel anchoring, which is an NHS-ester reaction between Acryloyl-X and amines in the proteins, at a pH value that is better adapted to NHS-ester chemistry (pH 8.3, in bicarbonate buffer, rather than neutral pH in PBS buffer, as generally performed in 4x expansion).
- 2) Increasing the stability of the proteinase K by removing EDTA from the buffers, and adding CaCl₂ (2 mM).

The anchoring and homogenization reactions worked far better, and the tissue could now be expanded. The expanded samples still present fractures (Appendix Fig. S5C), indicating a more problematic expansion than in the free-floating brain sections (albeit one should also take into account the fact that the slide-attached sections also present fractures before expansion; Appendix Fig. S5A). Nevertheless, the expansion of individual cells or tissue regions is still accurate (Appendix Fig. S5D,E).

To fully reply to the referee's comment, we have also added a discussion of the difficulties in using tissue slices in our main text (page 11).

- I am not clear exactly why the authors did not present confocal images of X10 samples. If the aim is to use a conventional microscope, that is ok. But still, a confocal stack would be very useful especially considering the presented 3D movies, which are mostly out of focus.

The expanded X10 samples are roughly 1000-fold dimmer than the original samples, due to the re-positioning of the fluorophores in the 1000-fold larger volume. At the same time, they are very large. Laser-scanning confocal imaging, which is still the most common type of confocal imaging, requires therefore long time periods, in which the sample is repeatedly scanned to obtain sufficient photons. One does remove here out-of-focus fluorescence, but the long imaging times allow for drift problems to come in, which are difficult to correct for.

We therefore decided to use conventional epifluorescence microscopy, which is faster, and allows the investigator to correct for any drift problems. At the same time, the fact that we performed Z-stacks enables us to use high-performance deconvolution software for eliminating the out-of-focus fluorescence. We now provide deconvolved movies for every one of our supplementary movies, which show that a good performance can be attained with this technique.

Nevertheless, confocal imaging is possible for X10, as we now show in the new Extended Version Figure EV4. Confocal imaging was reasonable for bright small structures, such as synapses (Fig. EV4A). This was not the case for dimmer protein clusters that are distributed on large organelles, such as TOM20 clusters on the surface of mitochondria. Here confocal imaging only revealed scattered dots, thus making the experiment difficult (Fig. EV4B), albeit the quality of the images is indeed high.

- Please add scale bars to all videos.

We have added the scale bars.

Referee #3:

Expansion microscopy achieves super-resolution capabilities by physical expansion of the tissue specimen and using conventional confocal or epifluorescence microscopes rather than sophisticated STORM or STED set-ups. Expansion microscopy techniques are easy to implement across Labs and have wide-spread biological applications as they can resolve closely apposed proteins and cell-compartments. In this EMBO report, Truckenbrodt et al describe a technique for a 10 fold expansion microscopy which is a 2.5 fold increment of the currently available 4 fold expansion microscopy procedures. This technique of x10 microscopy allows the authors to achieve a 25nm resolution, which is normally achievable only by STED or STORM techniques or by using the expansion microscopy procedure iteratively. The paper comes out of the Rizzoli Lab which is well-renowned in using sophisticated microscopy approaches when asking questions in neuroscience. The x10 method described in this report would be of interest to a wide-range of molecular and cellular biologists.

We thank the referee for the comments.

I have the following concerns:

1. There are caveats to this x10 approach as compared to the conventional 4x expansion microscopy method the most striking being that intrinsic fluorescent protein labels routinely being used in transgenic approaches, and which remains conserved in the 4x expansion procedure is lost in this x10 approach.

The referee is indeed right. We have mentioned this in the original manuscript. The only solution would be to immunostain the fluorescent proteins with antibodies, as suggested in the manuscript. Many laboratories have been successful with such immunostainings in the past, so we feel that this is not a major issue.

Further, considering the 10-fold separation between cellular compartments such as between pre- and postsynaptic compartments, one wonders if translinking protein interactions that normally span across the synapse are preserved with this x10 approach. Have the authors any data on whether labeling for a transsynaptic protein is possible with this x10 approach? The authors could either test for this if they can label for a transsynaptic protein with their method or they should speculate about this limitation to the applicability of their method in the discussion.

This is an important comment, and we have addressed it by immunostaining proteins whose trans-synaptic spacing is well known. We used two pairs of proteins, Bassoon and Homer, and RIM1/2 and PSD95. The average distance between the postsynaptic Homer and the presynaptic Bassoon across the synapse is 120-140 nm, as measured in STORM microscopy by Dani *et al.*, Neuron, 2010. The presynaptic RIM1/2 clusters are separated from the postsynaptic PSD95 by ~80 nm, according to the same study. We have measured these proteins in X10 microscopy, and found that the values were indeed well preserved (Fig. 4). The variation in the measurements, indicated in histograms in Fig. 4F, is also similar to that observed in STORM by Dani *et al.*, Neuron, 2010.

2. In Figure 4A the reader cannot see individual synaptic vesicles with the synaptophysin labeling- each synaptic vesicle should be about 40nm and thus well resolved by the authors x10 approach which allows a resolution of 25nm. Part of the problem might be that the images are taken with an epifluorescent scope and thus capturing the out of focus fluorescence. The authors should re-acquire the images with a conventional confocal microscope so the readers can better resolve the data in Figure 4 and also for the brain slices images shown in Figure 5. This should be easily addressable and is important to enhance the quality of data in Figure 4, Figure 5 and the associated suppl figures. If labeling with synaptophysin cannot allow clean punctate label then the authors should consider including data for presynaptic proteins that have more punctate expression such as vGlut.

As discussed for a comment from Referee #2 (page 3), confocal imaging is possible for X10 microscopy, but does not add much value. We have found the deconvolution of conventional microscopy Z-stacks to be much more efficient. We have now added deconvolved versions to all our movies.

At the same time, we have used the raw microscopy data from the brain slices to analyze the vesicle sizes (Appendix Fig. S5). We generated line scans over the Synaptophysin spots automatically, fitted them with Gaussian curves, and determined the resulting vesicle diameters. These peak at 50 nm, which is entirely in line with the vesicle sizes, taking into account that the original diameter of the vesicle (42 nm) is somewhat blurred by the antibody staining, due to the antibody size.

3. The images of Bassoon and Homer1 from hippocampal neurons in Figure 4 should include a pre-expansion image of the two synaptic markers beside the x10 expanded images. This will enable the reader to directly relate the effectiveness of the approach. A pre-expansion image of these markers is shown in Suppl Figure 6 but it is difficult to gather which field of view the supplemental image has been taken from. The main figure 4 itself should show pre- and post-expansion images of these synaptic markers for the same region/field of view.

We now show a gallery of pre- and post-expansion images for both Bassoon-Homer and RIM1/2-PSD95 pairs, to showcase this (Appendix Fig. S4).

As the individual synapses are very dense in our cultures, and are very small and punctate in the pre-expansion images, it is exceedingly difficult to find the same exact one after expansion. We typically image the pre-expansion samples with a low magnification objective, in which the synapses are not easy to differentiate from each other. This reduces the workload considerably, and keeps the sample bleaching to a minimum.

Imaging the same synapse with reasonable resolution both before and after expansion would involve imaging a large section of the coverslip with a high magnification objective, followed by finding the same synapse in the expanded gel. This is very labor-intensive, as the gels are ~20 cm in diameter, and need to be imaged with much care. Due to this, and to the fact that the first author moved during the revisions to a different laboratory, in a different country, we have not been able to accommodate this comment in time. However, we hope that Appendix Fig. S4 is sufficient for this purpose.

4. Figure 5C: It is unclear to the reader which part/region of the brain slice has been selected to show the three synaptic markers: please include an image of the three synaptic markers across the entire brain slice and place a box or a circle indicating the area represented in panel C.

We have only taken the images of the Bassoon and Homer at high resolution, since these are relatively dim in the brain samples, and we tried to avoid bleaching them by taking images at low resolution. The three-color image shown is from the boxed image in panel A, as we now indicate in the figure legend.

In addition, in response to referee #2, we have added a figure containing quantifications of the brain expansion, which provides more information on the expansion of brain slices. Please see the new Appendix Fig. S5.

5. Supplemental Figure 7: It is difficult for the reader to discern the synaptic markers in panel C and the associated movie - please remake the image stack with a confocal scope so the diffuse (green) signal can be better resolved.

We now provide the same exact movie in both raw and deconvolved form.

6. Supplemental Figure 7: No Homer signal is evident in panel B although the label suggests that there should be.

We have corrected this error. We are only showing two color channels in the particular figure panel (Synaptophysin and Bassoon), as the referee noticed.

7. While referring to the images from rat cerebellum the authors refer to 'the directional orientation of synapses'. It is unclear what is meant by this phrase and the authors should either clarify what they mean or exclude the use of such jargon.

We have now quantified the orientation of the synapses (Supplementary Fig. 10), and we now explain its meaning in the text. Briefly, we mean that many synapses in these brain areas have their active zones, the sites where they release the synaptic vesicles (which are marked by the Bassoon stainings), positioned at similar angles to the synapse centers. Such an arrangement does not take place in culture, for example, and is an interesting feature of the cerebellum sections we analyzed. The particular cerebellum area is dominated by so-called “parallel fibers”, which presumably position the synapses in a directional fashion. However, this is a minor point, which may not be of interest to the general reader, so we simplified this section considerably.

8. Similar concern for the phrase 'morphology of these structures' used in page 9 of the text. It is not directly apparent to the reader what is meant. Please clarify or omit.

We have now clarified this. We were referring to the overall organization of the Bassoon and Homer 1 immunostaining signals.

2nd Editorial Decision

4 June 2018

Thank you for the submission of your revised manuscript. We have now received the referee reports and I am happy to say that both referees support the publication of your paper now. We can therefore in principle accept it.

Only very minor changes are needed. The references may not list more than 10 author names. Please amend; the EMBO reports reference style is in EndNote.

Please add page numbers to the table of content of the Appendix.

I attach a word file with tracked changes by our data editors in the figure legends. Can you please make the necessary corrections using the track changes option and send the corrected word file back to us?

EMBO press papers are accompanied online by A) a short (1-2 sentences) summary of the findings and their significance, B) 2-3 bullet points highlighting key results and C) a synopsis image that is 550x200-400 pixels large (the height is variable). You can either show a model or key data in the synopsis image. Please note that text needs to be readable at the final size. Please send us this information along with the revised manuscript.

When uploading a new manuscript version in our online system you can bring forward all old manuscript files and then only replace the ones that need to be replaced.

I look forward to seeing the final version of your manuscript as soon as possible.

REFEREE REPORTS

Referee #2:

The authors satisfactorily addressed my concerns. I would suggest the publication of the work.

Referee #3:

The authors have addressed all my concerns. I have no further comments.

2nd Revision - authors' response

8 June 2018

Thank you for your comments on our manuscript entitled "*X10 Expansion Microscopy Enables 25 nm Resolution on Conventional Microscopes*".

We would also like to thank our reviewers for their thoughtful and constructive criticism.

We have performed all of the experiments that the referees suggested. As a result, we have included several new figures, which strengthen our manuscript considerably. We have also changed parts of our Results and Discussion, according to the suggestions of the referees.

We hope that our manuscript now is suitable for publication.

Corresponding Author Name: Silvio O. Rizzoli

Manuscript Number: EMBOR-2018-45836V2